# Head-Mounted and Hand-Held Displays Diminish the Effectiveness of Fall-Resisting Skills

**DOI:** 10.3390/s22010344

**Published:** 2022-01-04

**Authors:** Anika Weber, Julian Werth, Gaspar Epro, Daniel Friemert, Ulrich Hartmann, Yiannis Lambrianides, John Seeley, Peter Nickel, Kiros Karamanidis

**Affiliations:** 1Sport and Exercise Science Research Centre, School of Applied Sciences, London South Bank University, London SE1 0AA, UK; webera2@lsbu.ac.uk (A.W.); werthj2@lsbu.ac.uk (J.W.); g.epro@lsbu.ac.uk (G.E.); lambriy2@lsbu.ac.uk (Y.L.); seeleyj@lsbu.ac.uk (J.S.); 2Department of Mathematics and Technology, University of Applied Sciences Koblenz, 53424 Remagen, Germany; friemert@hs-koblenz.de (D.F.); hartmann@hs-koblenz.de (U.H.); 3Institute for Occupational Safety and Health of the German Social Accident Insurance (IFA), 53757 Sankt Augustin, Germany; Peter.Nickel@dguv.de

**Keywords:** smart glasses, head-mounted displays, hand-held displays, gait perturbation, stability control mechanisms, falls

## Abstract

Use of head-mounted displays (HMDs) and hand-held displays (HHDs) may affect the effectiveness of stability control mechanisms and impair resistance to falls. This study aimed to examine whether the ability to control stability during locomotion is diminished while using HMDs and HHDs. Fourteen healthy adults (21–46 years) were assessed under single-task (no display) and dual-task (spatial 2-n-back presented on the HMD or the HHD) conditions while performing various locomotor tasks. An optical motion capture system and two force plates were used to assess locomotor stability using an inverted pendulum model. For perturbed standing, 57% of the participants were not able to maintain stability by counter-rotation actions when using either display, compared to the single-task condition. Furthermore, around 80% of participants (dual-task) compared to 50% (single-task) showed a negative margin of stability (i.e., an unstable body configuration) during recovery for perturbed walking due to a diminished ability to increase their base of support effectively. However, no evidence was found for HMDs or HHDs affecting stability during unperturbed locomotion. In conclusion, additional cognitive resources required for dual-tasking, using either display, are suggested to result in delayed response execution for perturbed standing and walking, consequently diminishing participants’ ability to use stability control mechanisms effectively and increasing the risk of falls.

## 1. Introduction

In many industrial settings, such as logistics, order picking or manufacturing, continuous access to information and instructions is essential to perform work tasks. Apart from traditional methods, such as paper-based instruction, electronic displays are frequently used nowadays, and jobs using augmented or virtual reality tools are expected to increase globally from the current 2.6 million to 23.4 million by 2030 [1]. DHL International GmbH provides an example of this, having deployed smart glasses for order picking worldwide since 2019 [2]. Such glasses have various advantages, including speed of use, versatility and real-time access to contextual information [3]. Furthermore, their use or use of other head-mounted displays (HMDs) enables the worker to process and include contextual information while leaving the hands and arms free for use and maintaining a head-up posture. Thus, in work environments, such displays provide a promising alternative to the rather restricting paper lists or hand-held displays (HHDs) and increase efficiency and business turnover [4].

Despite their benefits, wearable technologies may have negative safety implications, especially when used during locomotion. If a cognitive task displayed on an HMD is processed during walking, the capability to perform both tasks will decrease [5]. Furthermore, smart glasses appear to induce conservative obstacle crossing strategies [6] and have been shown to negatively affect a person’s control of positioning in the lateral direction [7], which indicates a potential risk for falls. Whilst the causes of falls must be considered as multifactorial [8], divided attention, in dual-task situations when displays are used during walking, represents a significant potential hazard. This is of concern given that internationally about 20% of workplace accidents specifically relate to slips, trips and falls for level movement [9,10] and account for about 16% of fatal work accidents in the United States [11].

In addition to divided attention, dual tasks involve an increased cognitive load that can affect walking stability [12,13,14]. The redundancy of degrees of freedom (motor abundance) allows flexibility for the central nervous system to effectively control body movements, irrespective of environmental changes that are often challenging [15]. For walking itself (a single task), the central nervous system usually has enough cognitive resources to arrange for effective motor task outcomes. However, cognitive resources are limited and may be exhausted in a dual-task situation since they are divided between the motor and secondary tasks. This inevitably reduces the performance of at least one task [16]. If a secondary visual task is performed during walking, the dual-task costs are further amplified because of the difficulties of processing two visual streams simultaneously [17]. Accordingly, access of the central nervous system to multiple solutions to control walking decreases with the involvement of a secondary task, leading to reduced body stability and yet more so in the presence of external perturbations [18].

Motor behaviour relies on both reactive and predictive control to maintain postural stability under challenging conditions (e.g., sudden perturbations; [19]) and hence prevent falls. Investigation of fall prevention requires assessment of the ability to use certain stability control mechanisms after sudden perturbations [20]. According to Hof [21], stability loss can be counteracted by three fall-resisting mechanisms: (1) adjusting the application of force to the ground, e.g., by using a step to increase the base of support (BoS), (2) counter-rotating body segments around the centre of mass (CoM), e.g., moving arms and the trunk and (3) applying an external force, e.g., holding on to a handrail. Depending on the challenge to stability control and the environment (e.g., availability of space or handrails), different or combined mechanisms are required. For instance, the increase in the BoS is the most important mechanism for recovery from large perturbations and changes in the position of the CoM that occur during tripping [22,23]. Whether the use of augmented-reality HMDs or HHDs affect the execution of stability control mechanisms during perturbed standing and walking is currently unknown. Given the increased use of digital information via displays in industrial settings, there is a need to determine the effect of HMDs and HHDs on dynamic stability control to evaluate the risk of such use on falls for employees.

This study investigated the effect of HMDs and HHDs on stability control mechanisms for healthy adults involved in perturbed standing and walking tasks while simultaneously carrying out a cognitive task presented via a display by using an inverted pendulum model and analysing the components of the margin of stability (MoS). In addition, the effect of using each display on cognitive performance and physiological stress was examined by investigating cognitive response failure rate, heart rate variability and skin conductance response. It was hypothesised that the ability to effectively use mechanisms responsible for controlling stability after unexpected perturbations during standing and walking would be diminished when performing a cognitive task presented via an HMD or an HHD.

## 2. Materials and Methods

### 2.1. Participants

Fourteen healthy adults (age range 21 to 46 years; height 179 ± 9 cm; mass 76 ± 11 kg; physical activity 6 ± 2 h/week; means ± standard deviations) voluntarily participated in the present study after providing their consent in written form. Prior to participation, the participants were checked for inclusion criteria, i.e., normal or corrected-to-normal vision (e.g., glasses or contact lenses) and any known or diagnosed neurological and musculoskeletal impairments, via questionnaires to minimise the risk of impaired posture, gait or cognitive function during the investigation. The study was approved by the ethics committee of London South Bank University (approval number ETH2021-0018) and met all requirements for human experimentation in accordance with the Declaration of Helsinki [24].

### 2.2. Study Overview and Biomechanical Analysis

The participants were involved in two kinds of activity: single tasks, which were four main tasks in number involving standing and movement (Figure 1), and the respective dual tasks for which standing and movement were combined with a secondary cognitive task. For motor tasks, the abilities to both increase the BoS in the anterior direction and counter-rotate segments around the CoM were assessed. All motor tasks assessed stability along the anteroposterior axis. During all tasks, participants wore a safety harness connected to a ceiling-mounted system that prevented body contact with the floor (except for the feet). Participants viewed the cognitive task through head-mounted binocular smart glasses (HMD; HoloLens2, Microsoft, Washington, DC, USA) or using a tablet (HHD; Samsung Tab2, 0.6 kg, Samsung, Seoul, Korea) that was fixed with a strap to the left hand. To simulate the weight of rugged industrial tablets, additional weights were attached to the tablet to obtain a total weight of 1.3 kg. Unless otherwise specified, motor tasks were performed in counterbalanced order for HMD, HHD and CON (no display) conditions. The cognitive task was performed only for the HMD and HHD conditions. An overview of the motor tasks, display use and performance criteria is provided in Table 1. An optical motion capture system used 16 infrared cameras (120 Hz; Qualisys Track Manager v2019.3; Qualisys, Gothenburg, Sweden) to track the 3D coordinates of 9 retroreflective markers (placed on C7, left and right trochanter, malleolus lateralis, toe and heel) and assess CoM and BoS trajectories in the sagittal plane. To assess onsets of perturbations and analyse the participants’ stability control performances throughout the motor tasks, ground reaction forces were synchronously recorded using a TTL signal of the motion capture system by two force platforms (Kistler, 40 × 60 cm, Winterthur, Switzerland, 1080 Hz sample frequency) and analogue signals from two 1D strain gauge force transducers. Ground reaction force signals and 3D coordinates of the markers were smoothed using a recursive fourth-order Butterworth filter (cut-off frequency 20 Hz). Custom-built pneumatically driven devices were used to deliver perturbations (see the descriptions of perturbed standing and unperturbed and perturbed walking provided below). Analyses were computed using custom-made MATLAB routines (v2020b; MathWorks^®^, Natick, MA, USA). Finally, the physiological status of each participant was assessed via heart rate variability and skin conductance measurements and their perceived task load was registered using a questionnaire (National Aeronautics and Space Administration (NASA) Task Load Index, [25]).

### 2.3. Unperturbed Standing

This task tested the limits of stability for standing (Figure 1A). First, participants were instructed to adopt a quiet bipedal stance for 30 s on a force plate with their feet parallel and their heels 10 cm apart. They were then asked to lean forward gradually as far as possible and then return to the upright position without taking a compensatory step to regulate stability [26,27]. Afterward, participants performed the same task, but in the posterior direction. Unlike the studies mentioned earlier [26,27], no restrictions on hip, knee and ankle joint angles or arm movements were imposed on the return to upright stance. The centre of pressure (CoP) excursion path over time was assessed for the anteroposterior direction using ground reaction force data from the force plate. The limits of stability were calculated from the position of the CoP in relation to the anterior and posterior limits of the BoS (i.e., from toe and heel markers recorded using motion capture) to assess the individual’s boundaries of the BoS and were integrated in the analysis of the lean-and-release task as well as in the walking tasks (see the descriptions of lean-and-release task and unperturbed and perturbed walking provided below). Participants performed in both directions twice for each of the three conditions (CON, HMD and HHD), and the trial with the better performance (i.e., with the higher limits of stability) was used for further analysis of each condition. In addition, postural stability of quiet bipedal stance was evaluated from the total excursion distance of the CoP over a period of 25 s.

### 2.4. Perturbed Standing

This task assessed the use of counter-rotation of body segments around the CoM in response to perturbed standing (Figure 1B). As for the unperturbed standing conditions, participants stood on a force plate with their feet parallel and their heels 10 cm apart. They wore a chest strap connected via a Teflon cable to a software-controlled pneumatically driven perturbation device. After the participants were steady and performed no anteroposterior or mediolateral movements (checked in real time via the CoP), they were unexpectedly pulled in an anterior or posterior direction (separate tests by direction) via a Teflon cable (Figure 1B). Prior instruction asked participants to avoid taking a step to control stability. The initial perturbation magnitude was set at 10 N and 260 ms duration and was subsequently increased in 10 N increments (all at 260 ms) until the participants were unable to avoid a step to compensate for their postural instability for two consecutive trials. If the participants were able to maintain stability without a step at a perturbation magnitude of 200 N, the perturbation duration was gradually increased by 20 ms with the same pulling force (200 N). This combination of anterior and posterior perturbations (four trials in total) represented the CON condition. The perturbation magnitudes (anterior or posterior) that the participants were able to resist without taking a step under CON were applied subsequently to the HMD and HHD conditions (two times for each direction and display). The responses of the participants (stepping or no stepping) were recorded to assess the use of counter-rotations as a stability control mechanism. Please note, the input values for the pneumatic perturbation system (in bar) were calibrated to specific force outputs (in N) and its consistency with actual force outcomes was monitored using mechanical gauge sensors attached to the Teflon cable. However, more importantly, the perturbation magnitude during the dual-task conditions was matched with the magnitude during the control condition, revealing failure of stability control without taking a step.

### 2.5. Lean-and-Release Task

This task assessed the ability to effectively increase the BoS in response to unexpected stability loss from a static forward-lean position. Similar to previous studies [28], participants stood on a force plate, keeping their feet flat on the ground. They were placed in a forward-directed lean position and supported from behind via a horizontally running Teflon cable attached at one end to a belt worn at the pelvis level and at the other end to a custom-built pneumatically driven brake-and-release device (Figure 1C). The lean position was equivalent to 33% of the body mass and real-time controlled using a load cell placed in series with the Teflon cable. Once the participants were steady and performed no movements (i.e., anteroposterior or mediolateral mass displacements assessed via the load cell and CoP on the force plate), the cable was released unexpectedly within a time interval of 10 to 30 s [28]. The participants were encouraged to regain stability after release by a single recovery step that always landed on a second force plate mounted in front of the first one. Following three practice trials under CON, one trial for each condition (CON, HMD and HHD) was performed in counterbalanced order across the participants. 

To analyse the stability control performance, the anteroposterior MoS was calculated at touchdown of the first recovery step (ground reaction force data greater than 20 N) using a reduced kinematic model [29]. The MoS was defined as the difference between the anteroposterior boundaries of the BoS and the extrapolated CoM (XCoM; [30]). The CoM was calculated as the midpoint between the two pelvis markers (left and right trochanters). The velocity of the CoM (VCoM) was defined as the mean of the first-time derivatives of the CoM and C7 positions. XCoM was calculated as
XCoM=PCoM+VCoM/√(g/L)
where PCoM represents the CoM anteroposterior position, VCoM is the anteroposterior velocity of the CoM, g is gravitational acceleration (9.81 m/s^2^) and L is the reference leg length [29]. The BoS was corrected for the limits of stability (see information on unperturbed standing). Hence only the portion of BoS actively responsible for maintaining stability was considered. Finally, the rate of increase in BoS was calculated as the step length (i.e., toe off to touchdown of the first recovery step) divided by the step time. Toe off was determined using a foot-velocity algorithm according to Maiwald and colleagues (2009) [31] and touchdown from the point at which the ground reaction force exceeded 20 N.

### 2.6. Unperturbed and Perturbed Walking

This task simulated a trip-like situation to assess fall-resisting skills during locomotion. Firstly, participants walked back and forth along an 8 m section of flat, robust floor at a standardised speed of 1.4 m/s for approximately 3 min (C7 velocity was monitored in real time via the motion capture system). Following this, each participant’s ankles were connected via separate ankle straps and Teflon cables to a custom-built pneumatically driven brake-and-release device. MoS values averaged for three touchdowns during strap-attached unperturbed walking trials were used for further kinematic analyses. Please note that unpublished data from pilot testing revealed no differences in dynamic stability control for unperturbed walking with and without connection of the ankles to the brake-and-release device. At some point during the strap-attached walking, a restraining force generated by the brake-and-release device was suddenly applied and removed during the swing phase of the left leg (Figure 1D), causing an unexpected trip-like perturbation. The subsequent anterior increase in the BoS using the contralateral right leg was defined as the first recovery step. The MoS and components of dynamic stability (XCoM, BoS and rate of increase in BoS) at touchdown of the perturbed step (left leg) and at touchdown of the first recovery step (right leg) after simulated tripping were calculated using the methods described above (see information on lean-and-release task). However, as steps across all participants did not always land on a force plate during unperturbed and perturbed walking, touchdown was determined using kinematic data of the respective heel and toe markers [31]. MoS findings of perturbed walking were categorised as positive (i.e., stable body configuration at touchdown of the first recovery step) or negative (unstable body configuration). Participants were exposed to only one perturbation in order to analyse dynamic stability to a novel perturbation.

**Figure 1 sensors-22-00344-f001:**
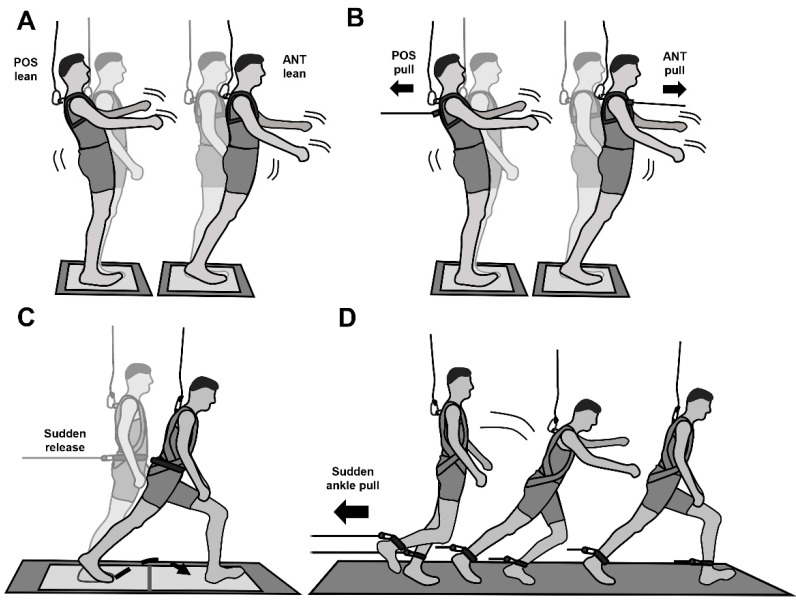
Illustration of analysed motor tasks. (**A**) Unperturbed standing: Assessment of the boundaries of the BoS (stability mechanism: counter-rotation). (**B**) Perturbed standing: Perturbation of standing in anterior and posterior directions with individually adjusted perturbation strengths (stability mechanism: counter-rotation). (**C**) Lean-and-release task with an inclination taking up 33% of body weight (stability mechanism: increasing BoS). (**D**) Perturbed walking: Simulated tripping while walking.

### 2.7. Cognitive Task

The secondary (dual) task was a spatial 2-back working memory task [32,33] presented to the participants using either an HMD or an HHD. For this task, a blue square changed its position randomly every 1.5 s within a 3 × 3 square matrix. If the blue square returned to the position where it had appeared two steps back (a target), the participants were instructed to respond by pressing a hand-held trigger (held in the hand not holding the HHD). If the blue square did not so return, the participant had to avoid responding. Participants were familiarised with the task while seated at a desktop computer prior to the main measurements.

The dual task was performed during all above-mentioned motor tasks to compare the effect of using either type of display on stability control. However, responses (i.e., trigger responses to correct targets in percent) were recorded only during unperturbed, quiet standing and unperturbed walking (for approximately 110 s per task) since other perturbation tasks were too short to evaluate the cognitive task.

### 2.8. Assessment of Physiological and Self-Perceived Stress

To investigate potential effects of using HMDs and HHDs on physiological stress, recordings were made of heart frequency (chest strap M400, Polar Electro Oy, Kempele, Finland) and skin conductance (finger sensor, eSense Skin Response, Mindfield^®^ Biosystems Ltd., Gronau, Germany). These were measured during both sitting and quiet standing while performing the cognitive task (approximately 110 s) for each condition (CON, HMD and HHD). Heart rate variability was calculated as the root mean square of consecutive R–R intervals (root-mean-square standard deviation; [34]). Skin conductance response was calculated as the difference between the maximum value during a measurement (CON, HMD and HHD) and the average value over 30 s recorded prior to the measurements (seated and without any other task performed; [35]). Neither parameter was measured during movement as physical activity would interfere the physiological responses. To compare self-perceived stress between the usage of an HMD and an HHD, a NASA Task Load Index (six questions; [25]) was handed out to the participants only after unperturbed, quiet standing and unperturbed walking under the respective dual-task conditions.

### 2.9. Statistics

All data were tested for normal distribution using the Kolmogorov–Smirnov–Lilliefors test (*p* > 0.05). To analyse possible effects on stability control parameters through the use of an HMD and an HHD compared to the CON condition, separate analyses of variance with repeated measures were performed (one-way ANOVA, condition as within-subject factor). These were carried out for all motor tasks as well as for the physiological stress parameters (heart rate variability and skin conductance response). Friedman tests for related samples were used in those cases for which normality assumptions were violated. Bonferroni post-hoc corrections (or Wilcoxon tests for related nonparametric samples) were computed to elaborate on the main effects of statistical analyses. Furthermore, the frequency of individuals in each group that showed negative MoS values (unstable body configuration) at touchdown of the first recovery step during perturbed walking was examined. To analyse for possible effects of unperturbed, quiet standing and unperturbed walking on cognitive performance (correct targets) and the NASA Task Load Index, related *t*-tests were separately performed for each display (HMD and HHD). Statistical significance was set at α = 0.05. Descriptive calculations and statistical analyses were performed using SPSS Statistics (v26, IBM; Chicago, IL, USA), MS Excel (v16.0, Microsoft Corporation, Redmond, WA, USA) and MATLAB (2020b, MathWorks^®^, Natick, MA, USA).

## 3. Results

The ability of the participants to maintain stability purely by counter-rotation, and without the need to perform a step, was assessed after externally induced anterior and posterior perturbations to bipedal standing. The perturbation magnitude while performing either of the dual tasks (HMD or HHD) was adjusted to the individual maximum at which participants did not step in the CON condition. Independent of use of an HMD or an HHD and the direction of perturbation, 57% of the analysed participants (in each dual-task condition) were not able to control stability without taking a step (Figure 2A). Analysis of the ability of participants to effectively increase the BoS after sudden anterior loss of stability (lean-and-release) revealed no significant effects of using either display on the main dynamic stability parameters (MoS, BoS and XCoM; Figure 3) and for rates of increase in BoS (CON 4.87 ± 0.43 m/s; HMD 4.76 ± 0.50 m/s; HHD 4.75 ± 0.57 m/s). No statistically significant main effects amongst conditions were found for total CoP excursion distance during quiet bipedal standing (CON 0.25 ± 0.07 m; HMD 0.27 ± 0.11 m; HHD 0.30 ± 0.10 m) or for the anterior and posterior limits of stability assessed during maximal forward and backward leaning (anterior CON 0.033 ± 0.007 m; HMD: 0.035 ± 0.007 m; HHD: 0.035 ± 0.01 m; posterior CON: 0.027 ± 0.007 m; HMD: 0.03 ± 0.01 m; HHD: 0.027 ± 0.006 m).

Concerning the perturbed walking task, 2 participants had been excluded due to marker artefacts and hence 12 participants were considered for the respective statistical analyses. Neither the use of an HMD nor the use of an HHD significantly differed from the CON condition in the effects on dynamic stability parameters (MoS, BoS and XCoM; Figure 4) and the rate of increase in BoS (CON 5.62 ± 0.76 m/s; HMD 6.09 ± 1.13 m/s; HHD 5.81 ± 0.69 m/s) for perturbed walking (simulated trips). However, individual scores with respect to stability at touchdown of the first recovery step showed negative MoS values for 75% (9 out of 12) and 83% (10 out of 12) of the participants for HMD and HHD, respectively, compared to 50% (6 out of 12) of the participants for CON (Figure 2B). Whilst four participants managed to regain stability in the CON condition (positive MoS values for first recovery touchdown), their MoS values were negative for one or both dual-task conditions (HMD or HHD) and hence further stepping actions were needed to regain stability. Within conditions (CON, HMD and HHD), the range in the BoS that was split by positive and negative MoS values was on average larger for HMD and HHD (0.22 and 0.10 m, respectively) than for the CON condition (0.05 m). There were no significant main effects regarding the MoS at touchdown of unperturbed walking (MoS CON: 0.05 ± 0.03 m; HMD: 0.05 ± 0.03 m; HHD: 0.06 ± 0.04 m).

Concerning the analysis of the cognitive task, the results revealed that the number of correct responses (targets) decreased from 91 ± 9% to 60 ± 17% (*p* = 0.003) and from 84 ± 12% to 55 ± 19% (*p* = 0.002) from quiet bipedal standing to unperturbed walking while using an HMD or an HHD, respectively. Furthermore, regarding the responses to the NASA task load self-assessment, significant differences were found in performance for the use of HMDs (*p* = 0.01) and HHDs (*p* = 0.03), decreasing for unperturbed walking compared to quiet bipedal standing (Table 2).

Analysis of the effect of using either display on physiological stress during sitting showed significant differences (*p* = 0.006) in heart rate variability for HMDs (42 ± 24 ms; *p* = 0.02) and HHDs (38 ± 21 ms; *p* = 0.007) compared to CON (68 ± 34 ms). For standing, there were no statistically significant differences in heart rate variability amongst HMDs (33 ± 17 ms), HHDs (32 ± 20 ms) and CON (32 ± 16 ms). Independent of the motor task (seated position and quiet bipedal standing), the skin conductance response was significantly higher (*p* ≤ 0.02) for both display conditions (HMD 2.8 ± 3.4 µA and 1.4 ± 1.4 µA; HHD 2.8 ± 2.6 µA and 1.5 ± 1.6 µA) compared to CON (0.7 ± 1.1 µA and 0.8 ± 0.7 µA).

## 4. Discussion

This study examined whether the use of an augmented-reality HMD or HHD diminishes a person’s ability to use stability control mechanisms in the form of counter-rotating body segments around the CoM and increasing the BoS. We could confirm our hypothesis that the ability to effectively use the mechanisms responsible for controlling stability, namely using counter-rotation of body segments around the CoM and increase in the BoS, after unexpected perturbations was diminished when processing a secondary cognitive task on an HMD and an HHD.

Both counter-rotating body segments around the CoM and increasing the BoS are important control mechanisms to regain stability after perturbations to locomotion [21,36]. In this study, the mechanisms of dynamic stability control during sudden, external postural perturbations were diminished while using both displays in comparison to the CON condition. Independent of the display used, 57% of the participants were not able to regain stability using counter-rotation during perturbed standing and had to increase their BoS by taking a step, even though the perturbation strength matched that determined in the CON condition. The main source of this diminished ability to use counter-rotation may be the additional cognitive resources required for the dual task [37]. Response execution is diminished, the XCoM moves outside the BoS and counter-rotation of segments around the CoM is not sufficient to regain stability. The participants must take an additional step as a result. Whilst a main stability control mechanism was clearly diminished in perturbed standing, we did not find any statistically significant differences in trip resisting skills (MoS and increase in BoS) for perturbed walking (Figure 4). However, when considering individual values, we found more participants showing an unstable body configuration (negative MoS) at touchdown of the first recovery step while using the HMD or the HHD (75% and 83%, respectively) compared to CON (50%; see Figure 2). Accordingly, those participants needed additional motor action to regain stability after the trip-like perturbation in order to avoid a fall (e.g., increasing the BoS by taking an additional step with the contralateral leg). Negative MoS in the dual-task conditions may be explained by lower individual values as well as higher ranges of the BoS when categorising by positive and negative MoS. These findings suggest that postural corrections are less effective when performing a dual-task using an HMD or an HHD [37], resulting in the reduced ability to respond effectively to a perturbation in tasks that require quick reactions (perturbed standing and perturbed walking in our case). In line with this suggestion, a previous study reported significantly higher response times and lower BoS in reactive stepping while performing a cognitively based dual task [38]. Our findings support the hypothesis that carrying out a cognitive task on either display diminishes the ability to counter-rotate and give an indication that the ability to increase effectively the BoS after sudden perturbations during locomotion is reduced.

While performing a dual task on both HMDs or HHDs diminished stability performance during perturbed standing and walking (simulated trip), the ability to increase the BoS to regain stability during the lean-and-release task was not significantly affected by using either display, i.e., the average range of the MoS at touchdown was similar for all conditions (0.160 to 0.166 m; Figure 3). Considering the MoS at the time of perturbation (i.e., touchdown of the perturbed step for perturbed walking and release for lean-and-release respectively), the values for perturbed walking were more negative by a factor of 2.6 than for the lean-and-release task (CON -0.42 m vs. −0.16 m; Figure 3 and Figure 4), indicating a higher task constraint for perturbed walking. The lower task constraint for the lean-and-release task is further supported by the observation that the average BoS and the average rate of increase in BoS were clearly lower in relation to perturbed walking (CON 0.82 m vs. 0.93 m; 4.87 m/s vs. 5.62 m/s). In addition, all participants reached a highly stable body configuration at touchdown of the first recovery step (MoS on average 0.16 m). Thus, given that the lean-and-release task was less demanding than perturbed walking, it seems possible that the cognitive resources of the analysed population of healthy adults were sufficient to effectively perform both cognitive and motor tasks.

Consistent with the results of a previous study, in which use of an HMD revealed no evident effects on walking performance compared to using a paper list [6], we found no statistically significant differences in CoP excursion path amongst conditions for unperturbed bipedal standing (mean values CON 0.25 m, HMD 0.27 m and HHD 0.30 m), limits of stability for maximal leaning (anterior CON 0.033 m, HMD 0.035 m and HHD 0.035 m; posterior CON 0.027 m, HMD 0.030 m and HHD 0.027 m) and dynamic stability control during unperturbed walking (CON 0.05 m, HMD 0.05 m and HHD 0.06 m). A possible explanation could be that participants have sufficient time to pre-plan their movements in predictive situations such as the motor tasks mentioned above. Despite this, both displays led to significantly lower self-perceived performance and significantly lower cognitive precision, resulting in approximately 25% fewer valid targets for walking compared to standing. Similarly, it has been reported that missed targets increased by approximately 20% for walking compared to those for sitting when completing a cognitive task with an HMD [5]. Participants must maintain paying continuous attention to the visual stimulus of the cognitive task, resulting in competition for resources between the cognitive task and visual guidance of participant gait [17]. The current results suggest that the participants prioritised their motor task (a posture-first strategy), with reduced response to the cognitive task for unperturbed walking compared to that for standing.

The influence of each display on cognitive and physiological demands under different dual-task pseudo-work conditions (standing, walking) was further investigated. Independent of the display used, it led to descriptively higher self-reported mental and temporal demand as well as higher effort and frustration for unperturbed walking in comparison to standing (NASA Task Load Index; see Table 2). The significantly lower self-reported performance matches with the measured lower performance in the cognitive task. Furthermore, heart rate variability decreased during sitting (mean values HMD 42 ms, HHD 38 ms and CON 68 ms) and skin conductance response increased during both sitting and standing for HMD and HHD compared to CON (HMD 2.8 and 1.4 µA, HHD 2.8 and 1.5 µA and CON 0.7 and 0.8 µA). Based on these results, it can be assumed that the physiological stress level of inexperienced users is increased on performing the cognitive task on either display compared to CON.

The findings of the current study are of practical relevance for health and safety at work in industry and services when using HMDs and HHDs. In unperturbed situations in the workplace, the influence on the dynamic stability control when performing a cognitive task with either display may be classified as a minor risk but as a significant risk for accidents and injuries in cases of unexpected postural challenges in stability due to trips, slips and falls. We found no differences between the two types of display for use of counter-rotation and increasing the BoS, as well as for cognitive performance and physiological stress. Based on the current protocol, it can therefore be assumed that neither display is preferable to the other in terms of fall risk. However, in situations where the hands must be used to apply external forces (e.g., holding onto a handrail to maintain balance), HMDs permit enhanced safety by allowing hand use, whereas this is not possible using HHDs.

It is worth mentioning that our participants were not experienced in using HMDs and HHDs. Therefore, the effect of divided attention due to dual tasking could be lower in more familiarised users, possibly resulting in an enhanced ability to use stability control mechanisms in perturbed situations. Furthermore, due to the current design, we cannot determine whether repeated perturbation trials and usage of the display conditions over time will lead to adaptive improvements in stability control mechanisms. Moreover, it is important to note that the cognitive performance could be measured during unperturbed standing and walking but due to the short perturbation time period, it was difficult to measure in perturbed locomotion. Future studies should investigate whether involving repeated perturbation training paradigms using HMDs and HHDs in visual dual tasking during standing and walking can enhance the ability to use stability control mechanisms to potentially reduce fall risk.

## 5. Conclusions

Our findings reveal that the ability to use fall-resisting skills in the form of counter-rotating body segments around the CoM was clearly diminished in perturbed standing and there was evidence of diminished ability to increase the BoS in perturbed walking. However, no evidence was found for head-mounted or hand-held displays affecting stability during non-strategic feedforward motor tasks such as unperturbed standing and walking. Additional cognitive resources required for dual tasking, using either type of display, are suggested to diminish the ability to use stability control mechanisms effectively for perturbed standing and walking, potentially increasing the risk of falls in these situations.

## Figures and Tables

**Figure 2 sensors-22-00344-f002:**
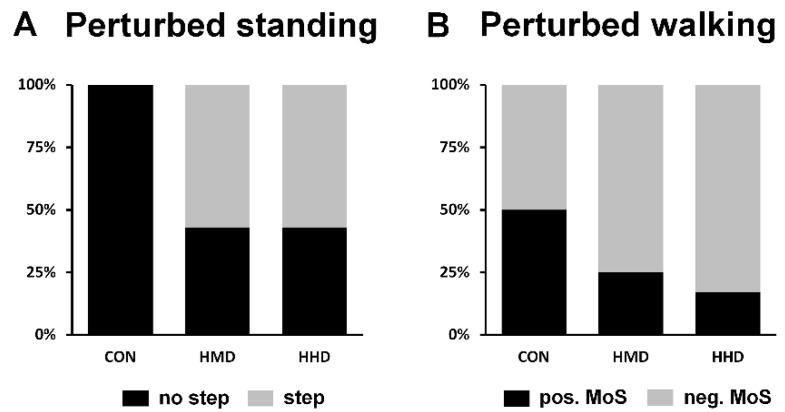
(**A**) Percentage of participants who required to regain balance in perturbed standing with and without taking a step for control condition (CON), head-mounted display (HMD) and hand-held display (HHD). (**B**) Percentage of participants with negative and positive MoS for CON, HMD and HHD for perturbed walking.

**Figure 3 sensors-22-00344-f003:**
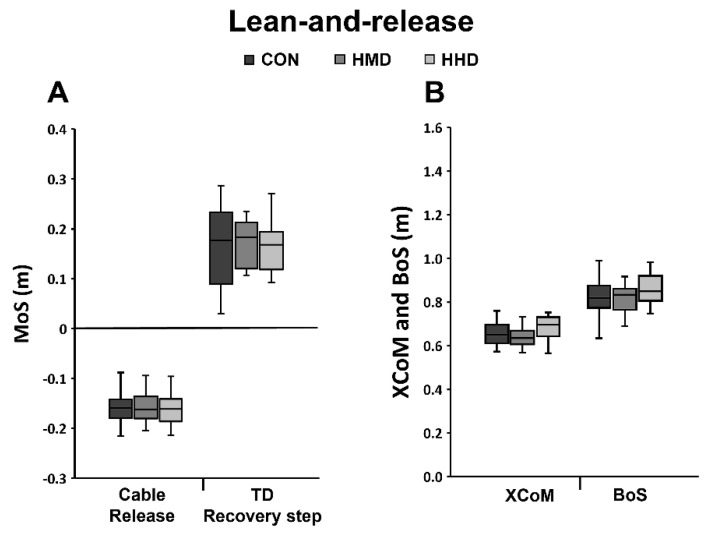
The margin of stability (MoS) and related stability components for the lean-and-release task illustrated with box plots (median, 1st and 3st quartile and whiskers) of all analysed participants for the control condition (CON), head-mounted display (HMD) and hand-held display (HHD). (**A**) MoS at the instants of release and at touchdown of the recovery step (TD); (**B**) extrapolated centre of mass (XCoM) and base of support (BoS) at TD of the recovery step.

**Figure 4 sensors-22-00344-f004:**
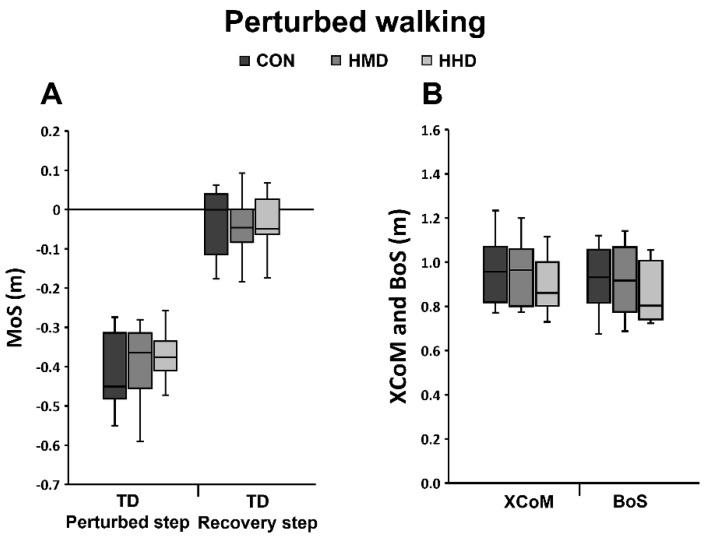
The margin of stability (MoS) and related stability components during unperturbed and perturbed walking illustrated with box plots (median, 1st and 3st quartile and whiskers) of all analysed participants for control condition (CON), head-mounted display (HMD) and hand-held display (HHD). (**A**) MoS at touchdown (TD) of the perturbed step and at touchdown of the first recovery step; (**B**) extrapolated centre of mass (XCoM) and base of support (BoS) at TD of the first recovery step.

**Table 1 sensors-22-00344-t001:** Study overview including type of motor task, type of display and cognitive task and the performance criteria. Each motor task was performed using three display conditions in counterbalanced order: head-mounted display (HMD), hand-held display (HHD) and no display (CON).

Study Overview		
Display and Cognitive Task	Motor Task	Outcome Parameters
HMD (Dual) HHD (Dual) CON (Single)	Sitting	Heart rate variability, skin conductance response
	Quiet bipedal standing	Heart rate variability, skin conductance response, NASA, cognitive task response, CoP total excursion distance
	Leaning	Limits of stability
	Perturbed standing	Stepping response
	Lean-and-release	MoS, BoS, XCoM, rate of increase in BoS
	Overground walking	NASA, cognitive task response, MoS, BoS, XCoM
	Perturbed walking	MoS, BoS, XCoM, rate of increase in BoS

*Note*: NASA: National Aeronautics and Space Administration Task Load Index; CoP: centre of pressure; MoS: margin of stability; BoS: base of support; XCoM: extrapolated centre of mass

**Table 2 sensors-22-00344-t002:** NASA Task Load Index items: mental, physical and temporal demand; performance; effort; and frustration in % for dual tasking on head-mounted displays (HMDs) and hand-held displays (HHDs) during standing and walking. * sig. different to standing within condition (*p* < 0.05).

	NASA Task Load (%)			
	HMD		HHD	
	Standing	Walking	Standing	Walking
Mental	48 ± 25	54 ± 26	44 ± 28	54 ± 23
Physical	17 ± 13	32 ± 24	33 ± 27	34 ± 23
Temporal	31 ± 22	50 ± 15	38 ± 27	43 ± 21
Performance	78 ± 19	48 ± 23 *	73 ± 21	53 ± 19 *
Effort	43 ± 28	59 ± 26	46 ± 28	55 ± 20
Frustration	22 ± 17	41 ± 30	23 ± 15	44 ± 21

## Data Availability

The datasets used and/or analysed during the current study are available from the corresponding author on reasonable request.

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
