# Peer review of "Head-Mounted and Hand-Held Displays Diminish the Effectiveness of Fall-Resisting Skills"

_sensors, 2022, doi:10.3390/s22010344_

Round 1

Reviewer 1 Report

Nice paper. Below I provide some remarks:

  1. I would like to have an abbreviation table, that would make reading easier.
  2. Formulation of the research needs a clear sentence - introduction finished by mentioning parameters rather then expected effect.
  3. There are unclear display use in the tests, reader addressed to read through paper text. I would recommend to create subchapter for explanation of situation in the terms of action of motoric and mental efforts as research situation and expected results.
  4. Discussion remains unclear - initial formulation will be echoed by results and their importance much better.

tfor

Author Response

Thank you for the fast review. Please see the attachment for a point-by-point response to your comments. Page and line numbers refer to text with track changes shown.

Reviewer 2 Report

The paper written by the following Authors: Anika Weber, Julian Werth, Gaspar Epro, Daniel Friemert, Ulrich Hartmann, Yiannis Lambrianides, John Seeley, Peter Nickel and Kiros Karamanidis, entitled “Head-mounted and hand-held displays diminish effectiveness 2 of fall-resisting skills” presents an interesting study on an application of head-mounted and hand-held displays for the stability control mechanisms and impair resistance to falls.

Although the paper is interesting, I have some major concerns:

Title

The title reflects the results presented here.

Abstract

  1. The abstract is lacking the aim of the study, material and methods description as well as an informative conclusion. It should be written in more details.

Material and Methods

  1. What does it mean “They had normal or corrected-to-normal vision and were free of neurological and musculoskeletal impairments that might have affected posture, gait or cognitive function”? Did Authors perform any medical study?
  2. How the force value was measured and determined?
  3. How many repetitions were performed for each test?

Discussion

  1. It would be better if conclusions were a separate chapter.
  2. In the discussion part there is no limitation to the studies. It should be included in the manuscript.

Author Response

(The authors gave the same response as above.)

Round 2

Reviewer 2 Report

I accept the manuscript in the present form.